# Revolutionizing Cancer Research: The Impact of Artificial Intelligence in Digital Biobanking

**DOI:** 10.3390/jpm13091390

**Published:** 2023-09-16

**Authors:** Chiara Frascarelli, Giuseppina Bonizzi, Camilla Rosella Musico, Eltjona Mane, Cristina Cassi, Elena Guerini Rocco, Annarosa Farina, Aldo Scarpa, Rita Lawlor, Luca Reggiani Bonetti, Stefania Caramaschi, Albino Eccher, Stefano Marletta, Nicola Fusco

**Affiliations:** 1Division of Pathology, IEO, European Institute of Oncology IRCCS, 20139 Milan, Italy; chiara.frascarelli@ieo.it (C.F.); eltjona.mane@ieo.it (E.M.); elena.guerinirocco@ieo.it (E.G.R.); nicola.fusco@ieo.it (N.F.); 2Department of Oncology and Hemato-Oncology, University of Milan, 20122 Milan, Italy; 3Biobank for Translational and Digital Medicine, IEO, European Institute of Oncology IRCCS, 20139 Milan, Italy; giuseppina.bonizzi@ieo.it (G.B.); camilla.rosellamusico@ieo.it (C.R.M.); cristina.cassi@ieo.it (C.C.); 4Central Information Systems and Technology Directorate, IEO, European Institute of Oncology IRCCS, 20139 Milan, Italy; annarosa.farina@ieo.it; 5Department of Diagnostics and Public Health, Section of Pathology, University of Verona, 37134 Verona, Italy; aldo.scarpa@univr.it (A.S.); stefano.marletta92@gmail.com (S.M.); 6ARC-Net Research Centre and Department of Diagnostics and Public Health, University of Verona, 37134 Verona, Italy; rita.lawlor@arc-net.it; 7Section of Pathology, Department of Medical and Surgical Sciences for Children and Adults, University of Modena and Reggio Emilia, University Hospital of Modena, 41121 Modena, Italy; luca.reggianibonetti@unimore.it (L.R.B.); stefania.caramaschi@unimore.it (S.C.); 8Division of Pathology, Humanitas Cancer Center, 95045 Catania, Italy

**Keywords:** biobank, digital pathology, cancer research, artificial intelligence

## Abstract

Background. Biobanks are vital research infrastructures aiming to collect, process, store, and distribute biological specimens along with associated data in an organized and governed manner. Exploiting diverse datasets produced by the biobanks and the downstream research from various sources and integrating bioinformatics and “omics” data has proven instrumental in advancing research such as cancer research. Biobanks offer different types of biological samples matched with rich datasets comprising clinicopathologic information. As digital pathology and artificial intelligence (AI) have entered the precision medicine arena, biobanks are progressively transitioning from mere biorepositories to integrated computational databanks. Consequently, the application of AI and machine learning on these biobank datasets holds huge potential to profoundly impact cancer research. Methods. In this paper, we explore how AI and machine learning can respond to the digital evolution of biobanks with flexibility, solutions, and effective services. We look at the different data that ranges from specimen-related data, including digital images, patient health records and downstream genetic/genomic data and resulting “Big Data” and the analytic approaches used for analysis. Results. These cutting-edge technologies can address the challenges faced by translational and clinical research, enhancing their capabilities in data management, analysis, and interpretation. By leveraging AI, biobanks can unlock valuable insights from their vast repositories, enabling the identification of novel biomarkers, prediction of treatment responses, and ultimately facilitating the development of personalized cancer therapies. Conclusions. The integration of biobanking with AI has the potential not only to expand the current understanding of cancer biology but also to pave the way for more precise, patient-centric healthcare strategies.

## 1. Introduction

In the last few decades, biobanking has emerged as a crucial and indispensable field in cancer research [1]. It has gained increasing prominence due to its providing a priceless resource—organized collections of biological samples and associated data—which serve as the bedrock for comprehensive analysis and the execution of clinical trials [1,2].

The scientific community has witnessed a remarkable surge in the utilization of human biological samples and associated data derived from biobanks [3]. This trend is substantiated by a growing body of scientific literature that underscores the significance of these biobanks in advancing our understanding of cancer [3,4,5,6,7]. As such, biobanks house an expansive array of diverse datasets sourced from various origins, encompassing a broad spectrum of bioinformatics and “omics” sciences data [8]. This rich tapestry of information harbors the immense potential to drive significant advancements in cancer [2,8].

At their core, biobanks offer datasets that extend beyond mere patient disease information. Within their repositories lies a treasure trove of biological samples, ranging from tissue specimens to various types of biofluids, all meticulously accompanied by annotated pathological data [9]. Nevertheless, despite the substantial promise that has arisen since the inception of biobanks and their subsequent regulation, these critical research infrastructures have grappled with various constraints and challenges, chiefly revolving around data acquisition, storage, and operational protocols [10,11,12]. This overarching dilemma has historically crystallized into the central quandary biobanks face: the effective utilization of the meticulously stored samples and associated data.

Despite the vast amount of data that can be collected, the optimization of this wealth of information becomes essential. Such challenges are intrinsic to the process of collecting and analyzing biobank data [13,14,15]. More recently, the integration of mathematical algorithms and machine learning technologies has emerged as promising tools to overcome these obstacles. As precision medicine increasingly leverages artificial intelligence (AI) techniques, biobanks are evolving from mere repositories of biospecimen collections to integrated computer datasets [2]. This transformation is akin to the metamorphosis of biobanks into dynamic, data-driven hubs that facilitate a deeper understanding of cancer biology. The interplay between AI and machine learning applications and biobanking in cancer research is becoming increasingly evident, with both domains mutually benefiting [16] (Figure 1). AI and machine learning offer the capacity to extract valuable insights from vast datasets that would be challenging, if not impossible, for humans to analyze comprehensively. These technologies can uncover intricate patterns, relationships, and predictive markers hidden within the data, which can significantly impact our understanding of cancer and its myriad facets. Furthermore, they have the potential to enhance the accuracy and efficiency of diagnosing and treating cancer patients, ushering in an era of personalized medicine tailored to individual genetic profiles and disease characteristics.

This paper aims to delve into the pivotal role of Digital Biobanks in the era of digital pathology, highlighting the incorporation of artificial intelligence in this field. Additionally, it seeks to explore the prospects and potentials of AI and machine learning applications within biobanking, particularly in cancer research.

## 2. Next-Generation Biobanking: Transitioning from Traditional to Digital Platforms

Biobanks serve as comprehensive repositories for varying types of biological samples and their associated data. Their primary objective is to process and preserve biospecimens according to strict criteria and provide researchers with essential resources for translational research and clinical investigations [2,16] and thus improve clinical outcomes and health care. The range of samples that can be collected and processed by biobanks is extensive, encompassing tissues (both normal and tumor), biofluids (such as blood, serum, plasma, urine, saliva, cerebrospinal fluid, effusion, bone marrow fluid, sperm, and cord blood), stool, purified cells from various tissues, peripheral blood cells (PBCs), and nucleotides (including DNA, RNA, and miRNA) among others [3]. These biospecimens are integrated with associated data, such as clinicopathological information, genetic profiles, medical record data, lifestyle data, and personal details. Consequently, it is imperative to augment the samples with up-to-date, pertinent data [4].

First, biobanks were created to preserve biospecimens over time [17,18]. To facilitate this objective, biological samples are typically preserved within cryogenic facilities, encompassing specialized refrigerators or warehouses to facilitate increased potential use of the samples. Additionally, biobanks may store matched formalin-fixed paraffin-embedded (FFPE) tissues, which provides samples that can be used for validation studies as these are the sample types used in clinical routine. It is worth noting that biobanks collect tissues from residual surgical specimens and biopsies obtained through minor surgical procedures (e.g., ultrasound-guided biopsies) [19]. The latter may require specific collection as their quantity is limited and processing is immediate. Both surgical samples and biopsies require involvement with the clinical diagnostic process. All the biological samples and associated data gathered must adhere to well-defined standard operating procedures. They should always be obtained with the patient’s informed consent, demonstrated by signing a research agreement or an equivalent form [19]. It is not always possible for a biobank to identify the specific research uses of the samples and data during the collection process. Therefore, different levels of consent are employed by biobanks, which differ in the levels of interaction with the patient for each potential use. A modern biobank should be able to engage with various stakeholders, such as research groups, clinical units, political institutions, biotech companies, and the pharmaceutical industry [20]. Whenever a scientific project needs biobanking support, it becomes vital to provide well-preserved samples and data that conform to prescribed requirements. The data must be securely stored, readily accessible, and traceable to effectively manage multiple projects concurrently. Furthermore, another crucial facet of biobanking management is the implementation of an integrated Laboratory Information Management System (LIMS) software that can integrate clinical records and patient data in addition to registering all samples-related data and sample processing data [21,22,23]. To secure this integration, it is critical to obtain the informed consent of all patients for legal and ethical requirements [19].

### 2.1. Precision Medicine

Precision medicine, a field rooted in analysing samples accompanied by clinical data, relies heavily on a vast amount of information to stratify patient treatment. This necessity is linked to the often weak connections between cancer phenotypes and clinical variables [24]. Consequently, in precision medicine research, “big data analytics” and “AI” have become indispensable. To expedite scientific advancements and maximize their impact on healthcare, the availability of well-characterized, high-quality samples and associated data in biobanks is paramount [25]. Addressing these demands requires a vast capacity and robust informatics capabilities. As medical researchers expand their horizons, their appetite for data keeps growing, leading to the collection and management of extensive sample series from diverse sources. The generic term “Big Data” refers to extremely large complex sets of data produced in a cancer research context, typically from patient health records, cancer registries, and large-scale genomic and genetic sequencing, the latter of which avail of biobank samples which guarantee the uniformity and reproducibility of results. Analysis of this “Big Data” encompasses innovative computational technologies and software that facilitate knowledge extraction from broad and heterogeneous datasets, including biological and medical information, thereby translating it into actionable insights [26]. This rapid acquisition, discovery, and processing are achieved through the application of computational mechanisms. Consequently, biobanking is directly affected by the paradigm shift demanded by Big Data, encompassing alterations in storage requirements and data analysis approaches.

### 2.2. The Role of AI

AI refers to computer technologies that simulate human intelligence, including cognitive abilities, deep learning, adaptability, engagement, and sensory comprehension [27,28]. Some systems can perform tasks usually requiring human interpretation and judgment in a significantly reduced timeframe [29]. These methods find applications across various fields, especially in health and medicine. The integration of AI in medicine can be dated back to the 1950s when physicians first employed computer programs to enhance diagnostic capabilities [30,31]. However, recent advancements in computational power and the availability of digital data have propelled a surge of interest and progress in medical AI applications [32].

These considerations notwithstanding, the practice of medicine is undergoing gradual transformation due to the influence of AI, impacting a wide range of medical disciplines, such as clinical, diagnostic, rehabilitative, surgical, and prognostic procedures. AI technologies now play a key role in disease diagnosis and clinical decision-making, leveraging vast amounts of data from diverse sources to detect illnesses and make informed judgments [33]. Thanks to the generation of huge amounts of medical big data, AI algorithms can effectively process and reveal novel insights that would otherwise remain hidden [34,35,36]. Therefore, these technologies facilitate the discovery of new drugs, improve healthcare services and enhance patient care [30,31].

Unlike traditional computer algorithms that follow predefined rules, AI systems learn and adapt through exposure to training data [37]. This learning process enhances the accuracy and reliability of AI prediction models. However, implementing new technologies in healthcare also raises concerns about potential inaccuracies and data privacy concerns. Inaccuracies can occur in the AI models based on the digital images and data used to train the models and may include those introduced through digital image staining variation. Data variation from different sites used to provide the basis for AI modelling will also affect the accuracy of the AI algorithms. Given the critical nature of healthcare-related data, collaboration between AI systems and physicians is essential [38]. To ensure the responsible integration of AI solutions, a robust governance framework is necessary to protect patients from harm and unethical behavior [39].

Ethical considerations are paramount in AI implementation, as biases can be incorporated into models through improper data collection or usage methods [40]. These biases can be many but are fundamentally based on the cohort makeup and data used in the specific AI systems. There are no established guidelines or standards for reporting and comparing AI models. It will be essential to specify these potential biases introduced in the developed systems, and therefore, future research should address this gap to guide researchers and clinicians [41]. As AI becomes increasingly indispensable in modern digital systems, ensuring ethical decision-making is free from unfair biases is crucial. Responsible AI systems should be transparent, explainable, and accountable [42]. This is of particular concern, given that they have the potential to enhance patients’ management and surgical outcomes and complement or replace existing systems. Neglecting the use of AI in healthcare may be considered unscientific and unethical. Life sciences, molecular biology, biotechnology, and digital tools integration are driving a transformative revolution in society [43]. These novel technologies enable personalized and preventive medical approaches, addressing challenges such as demographic change, healthcare accessibility, and sustainability. In oncology, the utilization of biomarkers and digital pathology through Digital Biobanks facilitates personalized diagnostic-therapeutic approaches, improving patient outcomes and resource management [21]. Integrating AI and digital pathology enhances the speed, accuracy, and remote capabilities of pathology diagnostics relying on novel small devices [44,45], enabling multidisciplinary consultations and supporting clinical trials. AI implementation in biobanks can identify new biomarkers, develop diagnostic strategies, and provide support in the selection of targeted therapies, ultimately leading to environmentally friendly hospital care with reduced costs and improved efficiency [46]. That said, AI requires the expertise of medical professionals to qualify the data and, particularly the digital images, to inform and train the AI algorithms. The future, thus, lies in the collaboration between AI and medical professionals, maximizing their combined strengths to improve healthcare outcomes.

### 2.3. Biobank Quality

To ensure the quality of samples and procedures, biobanks follow international guidelines such as those provided by the U.S. National Cancer Institute and, more specifically, by best practices developed by the International Society for Biological and Environmental Repositories (ISBER) [47]. These are constantly updated as the biobanking discipline develops. Furthermore, the International Agency for Research on Cancer (IARC) have developed minimum technical standards and protocols for biobanks dedicated to cancer research through all processes to guarantee the quality of samples and data collected [48]. Biobank quality is further enhanced with appropriate accreditations such as those of the International Organization for Standardization (ISO) used in all research laboratories and healthcare scenarios. These have historically comprised ISO 9001:2015 specifically for quality management systems. ISO 17025:2017 covers the general requirements for the competence of testing and calibration laboratories, and ISO 15189:2022 for medical laboratories specifies the requirements for quality and competence. More recently, in recognition of the peculiarities of biobanking, a dedicated ISO has been developed, ISO 20387, which defines the general requirements for biobanking. This includes the requirements for operational quality management systems to guarantee consistent quality control and reproducible results [49,50].

### 2.4. Digital Biobank and SOPS

Digital Biobanks play a key role in the discovery and validation of biomarkers for disease diagnosis and treatment. These repositories, integrated with molecular diagnostic pathology, provide swift access to diverse sample collections and comprehensive clinical data. Traditional biobank samples are the milestone for implementing Digital Biobanks, which incorporate novel methodologies like liquid biopsies [51]. According to the International Agency for Research on Cancer (IARC), as precision medicine drives a shift toward data-centric approaches, biobanks now capture medical data in addition to downstream molecular data alongside well-characterized tissue samples [2]. The transformation into Digital Biobanks involves the digitization of pathology slides using high-throughput scanners, resulting in whole-slide imaging scans and the creation of imaging data banks. Digital slides in oncology enable remote pathology and oncology consultations, enhance diagnostic accuracy, and uncover embedded information [52]. These digitized slides undergo pathology-driven annotations to facilitate AI and deep learning algorithms [53]. Specialized equips of pathologists ensure consistency and reproducibility of the data. AI analysis of the collected data generates intelligent reports and identifies digital biomarkers, enhancing patient selection and treatment outcomes. This approach aims to standardize and harmonize predictive oncologic pathology while reducing costs and improving tests’ sensitivity and specificity [54]. The integration of Digital Biobanks and AI has the potential to revolutionize cancer patients’ care, refining clinical management protocols and personalized treatments.

### 2.5. Direct and Indirect Costs

Establishing and managing an effective and efficient biobank requires adequate resources, including physical infrastructure, instruments, software, reagents, and personnel. It also requires financial support to support the infrastructure, personnel and continued running costs. Personnel should include a team of professionals led by a biobank director, with roles for quality management, security management, patient education, sample collection, data collection, and registration including personnel to prepare samples for downstream studies and to acquire the digital images for the digital biobank [55]. It is also vital to ensure personnel to guarantee patient privacy through data protection. To guarantee sample quality, adequate cryogenic instrumentation, such as freezers or liquid nitrogen containers, is necessary for specimen freezing and storage. Labelers, barcode readers, and proper labeling systems ensure biomaterial identification and tracking [56]. Personal protective equipment and quality control instruments are required, along with fireproof cabinets for paper records. A computer system with hardware, software, and website integration for sample publication, including a LIMS, is essential. Access control measures and processes are required to guarantee the security of the samples and data. There must also be a disaster recovery plan to provide solutions to protect the samples and data in the event of man-made and natural disasters. Instruments such as those to create tissue microarrays can create high-quantity sample sets for validation studies. More recently, digital tools like high-resolution scanners support Digital Biobank initiatives, and these also require software tools and large data storage to archive these digital sample images. In disease-based biobanks such as cancer biobanks, often overlooked costs are those incurred in terms of hospital engagement in patient consent and sample collection, where they are integrated with the hospital process. In addition, the evaluation and collection of cancer samples within pathology require the involvement of pathology personnel who are not directly financed by the biobank and, as such, are an indirect cost. The total cost of establishing and maintaining a biobank varies based on the type of biobank and services provided, which affect the requirements for personnel, consumables, instruments, and desired quality. These considerations are vital for Digital Biobanks and the utilization of AI in research [55].

## 3. Applications, Challenges, and Opportunities of AI and Digital Biobanking

By more effective participant matching and recruiting, as well as more thorough data analysis, AI and machine learning have the potential to enhance, simplify, and accelerate clinical trials. By comparing historical data to the intended trial enrollment requirements, it could also be feasible to generate artificial control groups. Additionally, AI and machine learning may be utilized to forecast and comprehend potential adverse events and patient subpopulations more accurately [57]. The creation of “synthetic patients” by AI to resemble diagnostic or treatment results appears plausible. However, using AI and machine learning applications and treatments brings a set of uncertainties that must be addressed in clinical trial procedures and study reporting [58,59]. We intend to discuss developments at the intersection between AI and medicine in our dedicated series. A separate set of issues surround the assessment of progress. The criteria for evaluating and validating scientific research in medicine are well established in traditional clinical research when progress manifests itself as a novel treatment for a well-defined ailment [60]. The medical community expects the same degree of certainty when an AI and machine-learning algorithm is used as an intervention rather than a medicine. However, the criteria for characterizing and assessing artificial intelligence and machine-learning treatments have not been established yet [61]. If an implementation is regarded as the standard that will change, reform, and enhance clinical practice, what standards should be applied to AI and machine learning-based interventional research? Three key issues are generally to be addressed to figure this question out [62]. First, the study design must address a clinically relevant topic in a way that can affect the actions of the healthcare provider and enhance patient outcomes. Second, the intervention needs to be scalable, defined, and appropriate for the current purpose. It must produce results that can be applied to clinical issues with similar characteristics across a wide variety of demographics and disease prevalence and must not be impacted by variables beyond the scope of the problem. Do we require different criteria for this procedure, or can AI and machine learning-driven care fulfill the standards we demand from a revolutionary therapeutic project or lab-based diagnostic test? Third, the outcome must be advantageous for all patients under consideration, not simply those comparable to the ones with features and findings on whom the algorithm was trained. This is true when the study findings are used in a way that influences practice. This calls into question whether such algorithms should consider public health (i.e., the utilization of limited resources) when making recommendations for diagnostic or therapeutic measures and the extent to which such factors are employed in the decision-making process. Health experts and the general medical community have met with interest in such ethical issues for ages [63].

## 4. Conclusions

Life sciences, including molecular biology, biotechnology, and digital technology, are pivotal in the ongoing industrial and health revolution, reshaping society and unlocking unprecedented possibilities for personalized and preventive approaches to healthcare. Furthermore, ongoing technological progress nowadays also addresses issues regarding increased life expectancy, equitable healthcare access, and sustainability of the medical system itself.

Moreover, advances in molecular biology and biomarkers have revolutionized diagnostic inquiry in oncology, leading to precision diagnostic and therapeutic strategies thanks to collaboration among clinicians and laboratory technicians to promote effectiveness, appropriateness, and safety in cancer patient care. Laboratory medicine has thus assumed, and will continue to assume, a central and pivotal role in significantly improving the clinical approach through the various stages of a cancer patient’s life, from prevention to diagnosis, prognosis, and therapeutic monitoring. This approach represents a promising paradigm, affording patients enhanced prospects for receiving increasingly effective therapies while minimizing adverse events often related to non-targeted treatments. As a result, such comprehensive approaches yield clinical benefits for patients and optimize healthcare resource management prudently and sustainably. The advent of AI has increased the potential of precision medicine, providing the enhanced capability to analyze the enormous amounts of data produced in the genomic sequencing era and integrating the massive data sets that incorporate not only these genomic and genetic data but also all medical record data and digital imaging including radiology images and pathology images. Increasing numbers and types of data augment challenges for integration of these data as well as the risk for error, but biobanks that historically collect samples in a standardized fashion provide the key to guaranteeing that these data are also collected in an organized fashion. As such, they hold the key to ensuring reliable AI models. The synergy between AI and human expertise is essential for promising results. Many physicians look to technology as a catalyst for enabling medical professionals to work more efficiently by supporting their efforts, resulting in fewer errors and greater seamlessness. While there remain concerns regarding inaccuracies and biases in AI due to lack of governance and transparency, it is hoped that, in the near future, we will overcome these concerns regarding AI. AI regulations are being developed to address the issues regarding privacy, transparency, and governance, and these will help address and provide a way to adopt AI in healthcare with a more positive and thoughtful perspective.

## Figures and Tables

**Figure 1 jpm-13-01390-f001:**
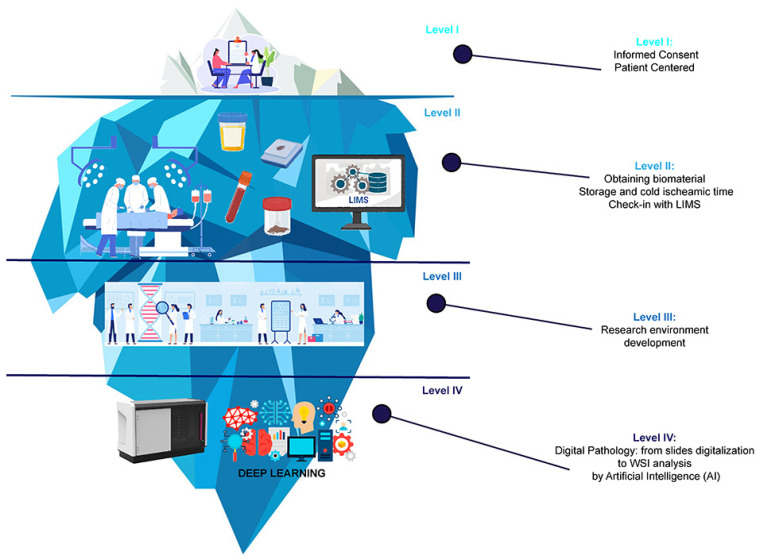
The journey of cancer research through the integration of Artificial Intelligence (AI) in the digital biobanking process. Schematic representation of the vast potential beneath the surface of conventional research methods, depicted as elements of an iceberg according to the various stages of digital biobanking. At the visible tip of the iceberg (Level I), we have the first crucial step of the process—patient-informed consent signing. Beneath the surface (Level II), the iceberg illustrates the extraction of tissue samples and biospecimens from cancer patients during medical procedures, as well as the biospecimen collection process and the implementation of sophisticated Laboratory Information Management Systems (LIMs). As we delve deeper into the iceberg (Level III), the illustration represents the development of the research environment. Eventually, at the bottom of the iceberg (Level IV) lies the digitalization of tissue slides by a digital scanner and the integration of deep learning, represented by complex neural networks used to analyze vast amounts of data derived from biospecimens, patient information, and other relevant research data.

## Data Availability

Not applicable.

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
