# Peer review of "Revolutionizing Cancer Research: The Impact of Artificial Intelligence in Digital Biobanking"

_jpm, 2023, doi:10.3390/jpm13091390_

Round 1

Reviewer 1 Report

Dear Editor,

Thanks for your invitation. Manuscript titled "AI-Driven Digital Biobanking: Transforming Cancer Research Paradigms". The purpose of this study is to highlight the potential uses of AI and machine learning in the field of biobanks and examine the crucial role that digital biobanks play in the era of digital pathology.

Minor corrections and explorations are needed to improve the quality of this review.

1.      The authors need to revise the manuscripts for language and grammar checks.

2.      The authors created a beautiful Fig., but the Fig. in the manuscript is not clear enough. The original or high-definition figures should be provided

3.      Scientific articles with several technical terms or acronyms should include an abbreviation list. Abbreviations can improve article reading and comprehension

Author Response

Reviewer #1:

Thanks for your invitation. Manuscript titled "AI-Driven Digital Biobanking: Transforming Cancer Research Paradigms". The purpose of this study is to highlight the potential uses of AI and machine learning in the field of biobanks and examine the crucial role that digital biobanks play in the era of digital pathology.

We thank the Reviewer for appreciating our manuscript.

Minor corrections and explorations are needed to improve the quality of this review.

  1. The authors need to revise the manuscripts for language and grammar checks.

We appreciate your input regarding the language and grammar of the manuscript. We have deeply revised and proofread the manuscript to ensure it meets the highest standards of language and grammar.

  1. The authors created a beautiful Fig., but the Fig. in the manuscript is not clear enough. The original or high-definition figures should be provided.

We are grateful for your positive feedback on the figure, and we understand your concern about its clarity. We will ensure that the original or high-definition figures are included in the submission to provide readers with a clearer visual representation.

  1. Scientific articles with several technical terms or acronyms should include an abbreviation list. Abbreviations can improve article reading and comprehension

Thank you for highlighting the importance of including an abbreviation list in scientific articles. We will coordinate with the journal’s production team to address this matter.

Reviewer 2 Report

The review's emphasis on the intersection of AI with biobanking, particularly in the realm of cancer research, is both timely and of paramount importance. By underscoring the transition from traditional biorepositories to agile, computational datasets, this work effectively chronicles the ongoing evolution in the domain. Overall, the presentation of the content appears sound, and I commend the authors for their efforts. However, for a more comprehensive and insightful read, I propose the following enhancements:

Given the burgeoning interest in the application of AI and ML to biobanks, the abstract seems to lack specific details regarding the methodologies employed. A concise mention of the underlying techniques, algorithms, or the nature of datasets under scrutiny—even in broad strokes—would enrich the abstract's depth.

While the conclusion aptly accentuates the game-changing potential of melding AI with biobanking in oncological studies, it could be augmented by briefly alluding to any encountered challenges or impediments during this integration. Additionally, insights or pointers for future endeavors in this direction would be a welcome addition.

I believe these modest amendments can elevate the review's clarity and utility for its readers.

Author Response

Reviewer #2

The review's emphasis on the intersection of AI with biobanking, particularly in the realm of cancer research, is both timely and of paramount importance. By underscoring the transition from traditional biorepositories to agile, computational datasets, this work effectively chronicles the ongoing evolution in the domain.  Overall, the presentation of the content appears sound, and I commend the authors for their efforts.

We thank the Reviewer for appreciating our manuscript.

However, for a more comprehensive and insightful read, I propose the following enhancements:

  1. Given the burgeoning interest in the application of AI and ML to biobanks, the abstract seems to lack specific details regarding the methodologies employed. A concise mention of the underlying techniques, algorithms, or the nature of datasets under scrutiny—even in broad strokes—would enrich the abstract's depth.

We appreciate your feedback on the abstract. We have incorporated more specific details regarding the methodologies, including the techniques, algorithms, and the nature of datasets used, even if briefly, to provide a clearer picture of the study's approach in the abstract.

  1. While the conclusion aptly accentuates the game-changing potential of melding AI with biobanking in oncological studies, it could be augmented by briefly alluding to any encountered challenges or impediments during this integration. Additionally, insights or pointers for future endeavors in this direction would be a welcome addition. I believe these modest amendments can elevate the review's clarity and utility for its readers.

Your suggestions to augment it by addressing challenges encountered during AI integration with biobanking in oncological studies and providing insights for future endeavors are well-received. We have made these adjustments to the conclusion, making it more informative and forward-looking, thus enhancing the overall clarity and utility of our review for our readers. Thank you for helping us improve our work.

Reviewer 3 Report

The manuscript raises interesting topic about using the artificial intelligence (AI) to search the repositories of biobanks. It is novel technology and it still needs improvements but it should be helpful to collect sets of samples with specific features for research projects.

Author Response

Reviewer #3

The manuscript raises interesting topic about using the artificial intelligence (AI) to search the repositories of biobanks. It is novel technology and it still needs improvements but it should be helpful to collect sets of samples with specific features for research projects.

We are immensely grateful to the Reviewer for the positive assessment of our work.

Once again, we would like to thank the Reviewers for their time and efforts dedicated to ameliorate our manuscript.